# I2Q: A Fully Decentralized Q-Learning Algorithm

**Jiechuan Jiang**
School of Computer Science
Peking University
jiechuan.jiang@pku.edu.cn

**Zongqing Lu**[†]
School of Computer Science
Peking University
zongqing.lu@pku.edu.cn

## Abstract

Fully decentralized multi-agent reinforcement learning has shown great potential for many real-world cooperative tasks, where the global information, *e.g.*, the actions of other agents, is not accessible. Although independent Q-learning is widely used for decentralized training, the transition probabilities are non-stationary since other agents are updating policies simultaneously, which leads to non-guaranteed convergence of independent Q-learning. To deal with non-stationarity, we first introduce stationary ideal transition probabilities, on which independent Q-learning could converge to the global optimum. Further, we propose a fully decentralized method, I2Q, which performs independent Q-learning on the modeled ideal transition function to reach the global optimum. The modeling of ideal transition function in I2Q is fully decentralized and independent from the learned policies of other agents, helping I2Q be free from non-stationarity and learn the optimal policy. Empirically, we show that I2Q can achieve remarkable improvement in a variety of cooperative multi-agent tasks.

## 1 Introduction

Multi-agent reinforcement learning (MARL) has shown great potential in real-world applications, including UAV [20], IoT [3], and games [29]. A number of MARL methods have been proposed for training agents to cooperatively maximize the cumulative shared reward, most of which follow the paradigm of centralized training and decentralized execution (CTDE), where the information of all agents is collected and used in the training phase. However, in many industrial applications where agents may belong to different companies, *e.g.*, autonomous vehicles or robots, the actions of other agents may not be accessible, so that CTDE methods cannot work. One way to address this challenge is *fully decentralized learning*, where the agents only use local experiences without the actions of other agents in both training and execution.

Independent Q-learning [28, 27] is one of the most straightforward decentralized methods. However, since other agents are treated as a part of the environment, from the perspective of an individual agent, the transition probabilities that also depend on the policies of other agents will change as other agents are updating their policies simultaneously [7]. Whether the agent updates the individual Q-values using the off-policy experiences, which are stored in the decentralized replay buffer, or the on-policy experiences, which are collected under the latest learned policies of all agents, the transition probabilities are non-stationary, thus the convergence of independent Q-learning is not theoretically guaranteed.

To tackle the non-stationarity problem, we propose a novel variant of independent Q-learning. First, we introduce *ideal transition probabilities* for each agent, which is induced by the optimal conditional joint policy of other agents, conditioned on action of this agent. We theoretically show that if all agents independently perform Q-learning on respective ideal transition probabilities, their policies

---

[†]Corresponding Author

36th Conference on Neural Information Processing Systems (NeurIPS 2022).

converge to the optimal joint policy when there is only one optimal joint policy. Certainly, such ideal transition probabilities are unknown in advance, but they can be deliberately learned by each agent in a decentralized way. We let each agent learn the QSS-value [6], the value of *state* and *next state*, which will converge on its replay buffer and be equivalent with the optimal joint Q-value in deterministic environments. Then, for each agent, the next state under ideal transition probabilities, given a state and an action, is the one with the highest QSS-value. Therefore, each agent can model an ideal transition function by learning QSS value and perform independent Q-learning on the ideal transition function, which guarantees the convergence to the optimal policy.

The proposed method, ***ideal independent Q-learning*** (**I2Q**), is fully decentralized, without the information of other agents. Although the theoretical proof is built on deterministic environments, I2Q can also be applied in stochastic environments. We theoretically analyze the value gap and experimentally demonstrate its effectiveness in stochastic environments. We evaluate I2Q on a variety of multi-agent cooperative tasks, *i.e.*, matrix games, MPE-based differential games [15], Multi-Agent MuJoCo [5], and SMAC [23], covering fully and partially observable, deterministic and stochastic, discrete and continuous environments. Empirically, I2Q outperforms baselines, verifying the analyzed convergence and optimality, and the modeled ideal transition function.

## 2    Related Work

**CTDE.** Most recent MARL methods follow the paradigm of centralized training and decentralized execution (CTDE). Policy-based methods [15, 8, 32, 37, 25, 18] extend policy-gradient into multi-agent cases by designing different optimization objectives. Value factorization methods [26, 22, 24, 31, 21] decompose the joint value function into individual value functions according to the Individual-Global-Max condition. In these methods, the information of all agents can be accessed in a centralized way during training, and the convergence can be guaranteed. Unlike these methods, we focus on decentralized learning where global information is not available.

**DTDE.** When agents cannot obtain global information, they have to adopt decentralized training and decentralized execution (DTDE). The most straightforward decentralized methods are independent Q-learning [27] and independent PPO [4]. However, since all agents are updating policies simultaneously, from the perspective of each agent, the transition probabilities are non-stationary, and thus the convergence of these methods may not be guaranteed. Fingerprints [7] uses iteration number and exploration rate to alleviate the problem of obsolete experiences in replay buffer for independent Q-learning. Hysteretic-QL [16] and Lenient-QL [17] let the agents be optimistic and attach less importance to the value punishment. However, none of these variants guarantees the convergence of independent Q-learning. Some theoretical methods [2, 38] guarantee the convergence to a Nash equilibrium, but the converged equilibrium may not be the optimal one when there are multiple equilibria [35]. Moreover, these methods cannot use replay buffer and have to re-collect experiences once the policies are updated, which is not practical. Our I2Q is a practical method to learn the optimal policy using replay buffer. Some studies [36, 11] consider decentralized learning with communication. However, in our fully decentralized settings, agents cannot share any information.

## 3    Method

We first introduce the problem setting and analyze the non-stationarity in independent Q-learning. Then, we construct ideal transition probabilities, on which the agents are theoretically guaranteed to converge to the optimal joint policy via independent Q-learning. Finally, we propose I2Q, which performs independent Q-learning on the modeled ideal transition function.

### 3.1    Preliminaries

There are $N$ agents in multi-agent MDP $M_{\text{env}} = <\mathcal{S}, \mathcal{O}, \mathcal{A}, R, P_{\text{env}}, \gamma>$ with the state space $\mathcal{S}$ and the joint action space $\mathcal{A}$. At each timestep, each agent $i$ performs an individual action $a_i$, and the environment transitions to the next state $s'$ by taking the joint action $\boldsymbol{a}$ with the transition probabilities $P_{\text{env}}(s'|s, \boldsymbol{a})$. For simplicity of theoretical analysis, we let all agents obtain the state $s$ [13], though in practice each agent $i$ may learn based on the observation $o_i \in \mathcal{O}$. All agents get a shared reward $r = R(s, s')$ and learn to maximize the expected return $\mathbb{E} \sum_{t=0}^{\infty} \gamma^t r_t$, where $\gamma$ is the discount factor. We consider the fully decentralized learning, where $M_{\text{env}}$ is partially observable to each agent since

each agent only observes its own action $a_i$ instead of the joint action $\boldsymbol{a}$. From the perspective of each agent $i$, there is an MDP $M_i = < \mathcal{S}, \mathcal{A}_i, R, P_i, \gamma >$ with the individual action space $\mathcal{A}_i$ and the transition probabilities

$$P_i\left(s'|s, a_i\right) = \sum_{\boldsymbol{a}_{-i}} P_{\text{env}}\left(s'|s, \boldsymbol{a}\right) \boldsymbol{\pi}_{-i}(\boldsymbol{a}_{-i}|s), \tag{1}$$

where $\boldsymbol{a}_{-i}$ and $\boldsymbol{\pi}_{-i}$ respectively denote the joint action and the joint policy of all agents except agent $i$. Each agent performs independent Q-learning:

$$Q_i(s, a_i) = \mathbb{E}_{P_i(s'|s, a_i)}\left[r + \gamma \max_{a_i'} Q_i(s', a_i')\right]. \tag{2}$$

In this paper, all $Q$ values denote the optimal values $Q^*$, we refer to $Q^*$ as $Q$ for simplicity. According to (1), the transition probabilities depend on the policies of other agents. Since other agents are updating their policies continuously, $P_i$ becomes non-stationary during training. Concretely, if each agent $i$ updates $Q_i$ in an off-policy way, using the experiences stored in the replay buffer $\mathcal{D}_i$, which does not contain other agents' actions, the transition probabilities $P_i$ in the replay buffer could be seen as the ones depending on the average policy $\bar{\boldsymbol{\pi}}_{-i}$ along the training process, which is non-stationary and outdated [7]. If each agent learns in an on-policy way, only using the most recent experiences, $P_i$ could be seen as the one under the latest learned policy $\boldsymbol{\pi}_{-i}$, which is not outdated but still non-stationary. The convergence of Q-learning on non-stationary transition probabilities is not guaranteed. *How to solve the non-stationarity?* In the next section, we introduce ideal transition probabilities and prove that if all agents perform independent Q-learning on the ideal transition probabilities, they will converge to the optimal joint policy.

## 3.2 Ideal Transition Probabilities

Assume there is only one optimal joint policy $\boldsymbol{\pi}^*(s) = \arg\max_{\boldsymbol{a}} Q(s, \boldsymbol{a})$, where $Q$ is the optimal joint Q-function,

$$Q(s, \boldsymbol{a}) = \mathbb{E}_{P_{\text{env}}(s'|s, \boldsymbol{a})}\left[r + \gamma \max_{\boldsymbol{a}'} Q(s', \boldsymbol{a}')\right]. \tag{3}$$

The unique optimal policy $\boldsymbol{\pi}^*(s)$ must be deterministic [19]. Thus, $\boldsymbol{\pi}^*$ can be uniquely factorized into deterministic optimal individual policies $\boldsymbol{\pi}^* = \langle \pi_1^*, \pi_2^*, \cdots, \pi_N^* \rangle$. For agent $i$, we build a deterministic optimal joint policy of other agents which is conditioned on $a_i$:

$$\boldsymbol{\pi}_{-i}^*(s, a_i) = \arg\max_{\boldsymbol{a}_{-i}} Q(s, a_i, \boldsymbol{a}_{-i}). \tag{4}$$

There may exist multiple $\boldsymbol{\pi}_{-i}^*$, and we can choose one arbitrarily. If other agents act the policy $\boldsymbol{\pi}_{-i}^*(s, a_i)$, the transition probabilities viewed by agent $i$ is $P_i\left(s'|s, a_i\right) = P_{\text{env}}\left(s'|s, a_i, \boldsymbol{\pi}_{-i}^*(s, a_i)\right)$, which are defined as ***ideal transition probabilities***. Then we have the following theorem.

**Theorem 1.** *If agent $i$ performs Q-learning on the ideal transition probabilities $P_i\left(s'|s, a_i\right) = P_{\text{env}}\left(s'|s, a_i, \boldsymbol{\pi}_{-i}^*(s, a_i)\right)$, its policy will converge to the optimal individual policy $\pi_i^*$.*

*Proof.* According to the fixed-point formulation of Q-learning, we have

$$Q_i(s, a_i) = \mathbb{E}_{P_{\text{env}}\left(s'|s, a_i, \boldsymbol{\pi}_{-i}^*(s, a_i)\right)}\left[r + \gamma \max_{a_i'} Q_i(s', a_i')\right], \tag{5}$$

$$\max_{\boldsymbol{a}_{-i}} Q(s, a_i, \boldsymbol{a}_{-i}) = \max_{\boldsymbol{a}_{-i}} \mathbb{E}_{P_{\text{env}}(s'|s, \boldsymbol{a})}\left[r + \gamma \max_{a_i'} \max_{\boldsymbol{a}_{-i}'} Q(s, a_i', \boldsymbol{a}_{-i}')\right] \tag{6}$$

$$= \mathbb{E}_{P_{\text{env}}\left(s'|s, a_i, \boldsymbol{\pi}_{-i}^*(s, a_i)\right)}\left[r + \gamma \max_{a_i'} \max_{\boldsymbol{a}_{-i}'} Q(s, a_i', \boldsymbol{a}_{-i}')\right], \tag{7}$$

where (6) is from taking $\max_{\boldsymbol{a}_{-i}}$ on both sides of (3), and (7) is by folding $\max_{\boldsymbol{a}_{-i}}$ into $P_{\text{env}}$. From (5) and (7), we see $Q_i(s, a_i)$ and $\max_{\boldsymbol{a}_{-i}} Q(s, a_i, \boldsymbol{a}_{-i})$ have the same stationary Bellman operator. According to the contraction mapping theorem, the Bellman operator converges to a unique fixed point. Thus,

$$Q_i(s, a_i) = \max_{\boldsymbol{a}_{-i}} Q(s, a_i, \boldsymbol{a}_{-i}).$$

Then, we have:
$$\max_{a_i} Q_i(s, a_i) = \max_{a_i} \max_{\boldsymbol{a}_{-i}} Q(s, a_i, \boldsymbol{a}_{-i}) = Q(s, \boldsymbol{\pi}^*(s)). \tag{8}$$

By contradiction, if an $\tilde{a}_i \neq \pi_i^*(s)$ but satisfies $Q_i(s, \tilde{a}_i) \geq Q_i(s, \pi_i^*(s)) = Q(s, \boldsymbol{\pi}^*(s))$, it contradicts that $\boldsymbol{\pi}^*$ is the unique optimal joint policy. Therefore, $\pi_i^*(s) = \arg\max Q_i(s, a_i)$. $\square$

By extending this theorem, if all agents perform Q-learning on such ideal transition probabilities under the conditional optimal joint policy, they will converge to the respective optimal individual policies, and arrive at the optimal joint policy $\langle \pi_1^*, \pi_2^*, \cdots, \pi_N^* \rangle$.

### 3.3 I2Q

*How to obtain the ideal transition probabilities from the non-stationary replay buffer for each agent?* We first introduce the QSS-learning [6]. Following the setting of QSS [6], we assume the environment is deterministic.[1] Each agent $i$ learns a value function $Q_i^{ss}(s, s')$ using the experiences in its own replay buffer $\mathcal{D}_i$:

$$Q_i^{ss}(s, s') = r + \gamma \max_{s'' \in \mathcal{N}(s')} Q_i^{ss}(s', s''), \tag{9}$$

where $\mathcal{N}(s')$ is the neighboring state set of the state $s'$ (the set of all next states of $s'$). The value function $Q_i^{ss}$ has several advantages. First, it has been proven in QSS [6] that $Q_i^{ss}$ and $Q$ learn equivalent values, which means

$$\max_{s'} Q_i^{ss}(s, s') = \max_{\boldsymbol{a}} Q(s, \boldsymbol{a}). \tag{10}$$

The equivalence shows that agent $i$ can independently infer the next state under the optimal joint policy according to $\arg\max_{s'} Q_i^{ss}(s, s')$. Second, as $Q_i^{ss}(s, s')$ decouples the action $a_i$ from the value and the implied transition probabilities: $P(s'|s, s') = 1, \forall s' \in \mathcal{N}(s)$, are always stationary, $Q_i^{ss}$ converges to the optimal value even on its own replay buffer, without the information of other agents. Third, since all agents act in the same environment, they collect the same state set and the same next state set in their own replay buffers. Therefore, they will converge to the same $Q_i^{ss}$ independently, which builds the consensus between the learned policies of agents.

*Then, how to build the ideal transition function from $Q_i^{ss}$?* As the environment $P_{env}$ and the optimal conditional joint policy of other agents $\boldsymbol{\pi}_{-i}^*(s, a_i)$ are deterministic, under the ideal transition probabilities $P_{env}(s'|s, a_i, \boldsymbol{\pi}_{-i}^*(s, a_i))$, the environment from the perspective of agent $i$ will deterministically transition to the next state that has the highest $Q_i^{ss}$ among all neighboring states given $a_i$:

$$s'^* = \arg\max_{s' \in \mathcal{N}(s, a_i)} Q_i^{ss}(s, s'), \tag{11}$$

where $\mathcal{N}(s, a_i)$ is the neighboring state set of $s$ given $a_i$. This is proved in the proof of the following theorem.

**Theorem 2.** *In deterministic environments, all agents will converge to the optimal policies, if each agent $i$ performs Q-learning on the transition function $s'^* = \arg\max_{s' \in \mathcal{N}(s, a_i)} Q_i^{ss}(s, s')$.*

*Proof.* The proof is given in Appendix A, which shows that $s'^* = \arg\max_{s' \in \mathcal{N}(s, a_i)} Q_i^{ss}(s, s')$ is the ideal transition function. $\square$

To model the stationary ideal transition function $s'^* = \arg\max_{s' \in \mathcal{N}(s, a_i)} Q_i^{ss}(s, s')$, we train a neural network $f_i(s, a_i)$ to predict $s'^*$ and update $f_i(s, a_i)$ by maximizing:

$$\mathbb{E}_{s, a_i, s' \sim \mathcal{D}_i} \left[ \lambda Q_i^{ss}(s, f_i(s, a_i)) - (f_i(s, a_i) - s')^2 \right]. \tag{12}$$

The first term enforces that the predicted next state has the highest $Q_i^{ss}$, and the second term constrains the predicted next state to be in the set $\mathcal{N}(s, a_i)$. The hyperparameter $\lambda$ is the coefficient. Based on the transition model $f_i(s, a_i)$, $Q_i^{ss}$ is updated by minimizing the TD-error:

$$\mathbb{E}_{s, a_i, s', r \sim \mathcal{D}_i} \left[ \left( Q_i^{ss}(s, s') - r - \gamma \bar{Q}_i^{ss}(s', f_i(s', a_i'^*)) \right)^2 \right], \quad a_i'^* = \arg\max_{a_i'} Q_i(s', a_i'). \tag{13}$$

---

[1] Although the environment $P_{env}$ is deterministic, from the perspective of each agent $i$, the viewed environment $P_i$ would still be non-stationary.

---
**Algorithm 1.** I2Q for each agent $i$
---
1: Initialize transition model $f_i$, Q-networks $Q_i$ and $Q_i^{\text{ss}}$, and the target networks $\bar{Q}_i$ and $\bar{Q}_i^{\text{ss}}$.
2: Initialize the replay buffer $\mathcal{D}_i$.
3: **for** $t = 1, \ldots, max\_iteration$ **do**
4:     All agents interact in the environment and store experiences $(s, a_i, s', r)$ in replay buffer $\mathcal{D}_i$.
5:     Sample a mini-batch from $\mathcal{D}_i$.
6:     Update $f_i$ by maximizing (12).
7:     Update $Q_i^{\text{ss}}$ by minimizing (13).
8:     Update $Q_i$ by minimizing (14) or (15).
9:     Update the target networks $\bar{Q}_i$ and $\bar{Q}_i^{\text{ss}}$.
10: **end for**
---

And $Q_i$ is updated by minimizing:

$$\mathbb{E}_{s,a_i,r\sim\mathcal{D}_i}\left[\left(Q_i\left(s,a_i\right)-r-\gamma\max_{a_i'}\bar{Q}_i(f_i(s,a_i),a_i')\right)^2\right].\tag{14}$$

$\bar{Q}$ is the target network of $Q$. When updating $Q_i$ (14), the target value is not computed on the next state sampled from replay buffer $D_i$, but on the one predicted by the modeled ideal transition function $f_i(s,a_i)$. In (14), the reward $r$ is simplified to depending just on $s$. When $r$ depends on both $s$ and $s'$, given $s$ and $a_i$, the next state under the ideal transition function (11) satisfies:

$$\begin{aligned}
Q_i^{\text{ss}}(s,s'^*) &= r(s,s'^*) + \gamma\max_{s''}Q_i^{\text{ss}}(s'^*,s'') \\
&= r(s,s'^*) + \gamma\max_{\boldsymbol{a}'}Q(s'^*,\boldsymbol{a}') \leftarrow (10) \\
&= r(s,s'^*) + \gamma\max_{a_i'}Q_i(s'^*,a_i') \leftarrow (8) \\
&= Q_i(s,a_i) \leftarrow (\text{deterministic ideal transition probabilities}).
\end{aligned}$$

Therefore, we can update $Q_i$ by minimizing:

$$\mathbb{E}_{s,a_i\sim\mathcal{D}_i}\left[\left(Q_i\left(s,a_i\right)-\bar{Q}_i^{\text{ss}}(s,f_i(s,a_i))\right)^2\right].\tag{15}$$

The training procedure of I2Q is summarize in Algorithm 1, where each agent $i$ learns $f_i$, $Q_i^{\text{ss}}$, and $Q_i$. Although all the modules update simultaneously, as the convergence of $Q_i^{\text{ss}}$ is guaranteed, the transition function $f_i$ derived from $Q_i^{\text{ss}}$ will be stationary in the later stage. Thus $Q_i$ will also converge. I2Q can be applied in environments with both discrete and continuous state-action space. In continuous action space, we build I2Q on DDPG [14], where a policy network $\pi_i(s)$ is trained by maximizing $Q_i(s,\pi_i(s))$ as a substitute of $\arg\max_{a_i}Q_i(s,a_i)$. In continuous state space, since $f_i(s,a_i)$ is differentiable, (12) can be maximized by gradient ascent. In discrete state space or large state space, we can map the state space to a continuous embedding space, and apply I2Q on the embedding space. We also provide an implementation without forward model $f_i$ in Appendix B.7.

### 3.4 Assumptions

We will further discuss the two assumptions. The prime one is that there is only one optimal joint policy, but I2Q can easily solve tasks with multiple optimal joint policies. With multiple optimal actions (with the max $Q_i(s,a_i)$), if each agent arbitrarily selects one of the optimal independent actions, the joint action might not be optimal. To address this, we can set a performance tolerance $\varepsilon$ and introduce a fixed randomly initialized reward function $\hat{r}(s,s')$ in the range $(0,\hat{r}_{\max}]$, where $\hat{r}_{\max}=(1-\gamma)\varepsilon$. Then all agents perform I2Q on the shaped reward $r+\hat{r}$ and learn the value function $\hat{Q}_i(s,a_i)$ in terms of $r+\hat{r}$. Since $\hat{r}$ is positive, $\hat{Q}_i(s,a_i)>Q_i(s,a_i)$. In $\hat{Q}_i(s,a_i)$, the maximal contribution from $\hat{r}$ is $\hat{r}_{\max}/(1-\gamma)=\varepsilon$, so the minimal contribution from $r$ is $\hat{Q}_i(s,a_i)-\varepsilon>Q_i(s,a_i)-\varepsilon$, which means that the maximal performance drop is $\varepsilon$ when selecting actions according to $\hat{Q}_i$. Moreover, since the reward function $\hat{r}(s,s')$ is randomly initialized, it is a small probability event to find multiple optimal joint policies on the reward function $r+\hat{r}$. Thus, if $\varepsilon$ is set to be small enough, I2Q could solve the task with multiple optimal joint policies.

The secondary one is deterministic environments. In stochastic environments, (10) does not hold, and $Q_i^{\text{ss}}(s,s')$ can be considered the "best possible value" [6]. So $s'^*=\arg\max_{s'\in\mathcal{N}(s,a_i)}Q_i^{\text{ss}}(s,s')$ is

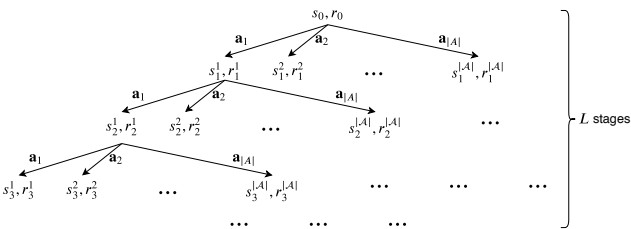

Figure 1: Illustration of the matrix games.

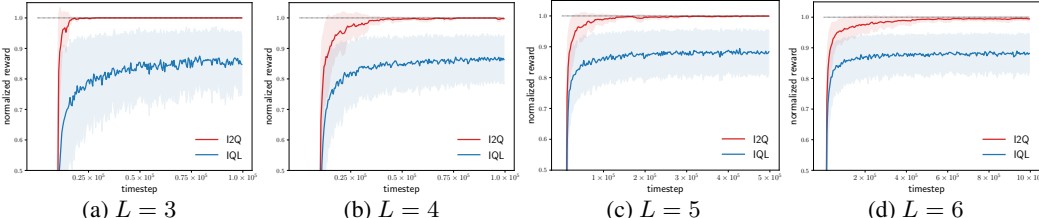

| (a) $L = 3$ | (b) $L = 4$ | (c) $L = 5$ | (d) $L = 6$ |

Figure 2: Learning curves on 100 random matrix games with different stage $L$.

the best possible transition function, rather than the ideal transition function. The difference between the learned $Q_i$ of I2Q and the true value under the ideal transition function $Q_i^{sto}$ is guaranteed by the following theorem.

**Theorem 3.** $\|Q_i^{sto} - Q_i\|_\infty \leq \frac{\Delta r}{1-\gamma} \left\| \frac{1 - P_{\text{env}}\left(s'^* | s, a_i, \boldsymbol{\pi}^*_{-i}(s, a_i)\right)}{1 - \gamma P_{\text{env}}\left(s'^* | s, a_i, \boldsymbol{\pi}^*_{-i}(s, a_i)\right)} \right\|_\infty$, *where* $\Delta r = r_{\max} - r_{\min}$.

*Proof.* The proof is given in Appendix A. $\qquad\square$

Theorem 3 shows that $Q_i$ of I2Q is closer to the true value if the transition probability of $s'^*$ is higher in stochastic environments. Since we sample transitions from $\mathcal{D}_i$ to update $f_i$ by maximizing (12), the second term of (12) makes the predicted next state $f_i(s, a_i)$ be close to the **high-frequency** next states in replay buffer $\mathcal{D}_i$ given $s$ and $a_i$, which means that the transition probability of the predicted next state $P_{\text{env}}\left(f_i(s, a_i) | s, a_i, \boldsymbol{\pi}^*_{-i}(s, a_i)\right)$ would not be too small. Thus, $Q_i$ of I2Q will be close to the true value and the worst cases where the predicted next states have very small transition probabilities can be avoided. *That is the reason why I2Q can be successfully applied in stochastic environments*

## 4 Experiments

In experiments, we first evaluate I2Q on a class of randomly generated matrix games to verify our theoretical analysis. Second, we compare I2Q with a series of Q-learning variants on MPE-based [15] differential games to illustrate the convergence and optimality of I2Q on more complex tasks with continuous action. Third, we test I2Q on two popular MARL benchmarks: Multi-Agent MuJoCo [18] and SMAC [23]. The experiments cover both fully and partially observable, deterministic and stochastic, discrete and continuous environments. Since we consider the fully decentralized settings, I2Q and the baselines *do not use parameter-sharing*. The results are presented using mean and standard deviation (std) with different random seeds. More details about hyperparameters are available in Appendix C.

### 4.1 Matrix Games

To support the theoretical analysis of I2Q, we perform experiments on a class of two-agent matrix games with $L$ stages. The action space of each agent is 3, so the joint action space $|\mathcal{A}| = 9$. The payoff tree is shown in Figure 1, where the reward $r$ is randomly generated in the range $[-1, 1]$ and fixed in each matrix game. Without loss of generality, we randomly generate 100 matrix games for each stage $L$, train the agents for four different seeds in each game, and plot the mean normalized return (normalized by the optimal return) and std over the 100 matrix games. Since the states are countable, we adopt Q-tables instead of neural networks, where $Q_i$ table and $Q_i^{ss}$ table are updating *simultaneously*, and use $\epsilon$-greedy policies. Figure 2 shows the learning curves with different stages $L$ under $\epsilon = 0.5$. IQL cannot converge to the optimal policies and shows large std, because it is afflicted

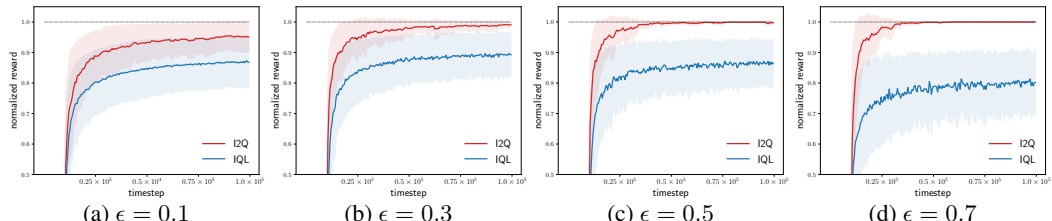

(a) $\epsilon = 0.1$      (b) $\epsilon = 0.3$      (c) $\epsilon = 0.5$      (d) $\epsilon = 0.7$

Figure 3: Learning curves with different $\epsilon$ on 100 random matrix games with stage $L$.

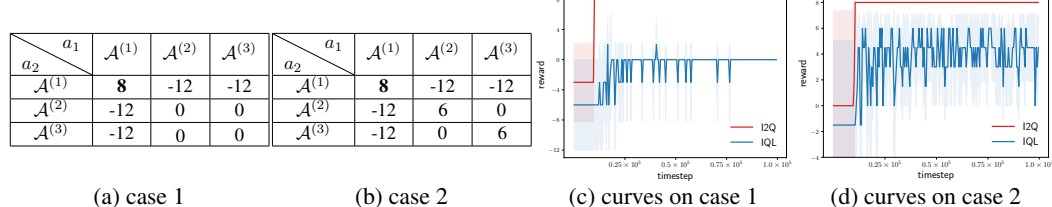

(a) case 1      (b) case 2      (c) curves on case 1      (d) curves on case 2

Figure 4: Learning curves on two specific one-stage matrix games.

with non-stationarity. I2Q converges to the global optimum on the matrix games, even when the stage $L$ grows, which confirms our theoretical analysis.

In Figure 3, we show the effect of exploration rate $\epsilon$ on the matrix games with $L = 4$. As $\epsilon$ increases, I2Q converges faster, while the performance of IQL drops. In conventional decentralized methods like IQL, there is a *dilemma of exploration and exploitation*. If $\epsilon$ is small, the weak exploration causes low sample efficiency and slow learning. When $\epsilon$ is large, all agents act more randomly. From the perspective of an individual agent, the transition probabilities will be much different from the real ones under the learned policies of other agents, which causes bad performance. That is the reason why IQL drops when $\epsilon$ increases. I2Q can avoid the dilemma, because $Q_i^{\mathrm{ss}}(s, s')$ is independent from agent policies and converges faster with larger $\epsilon$ as it sees more states. Thus, I2Q can use large $\epsilon$, *e.g.*, $0.5$ or $0.7$, to promote sample efficiency.

Moreover, we test I2Q on two one-stage matrix games proposed in QTRAN [24] and QPLEX [31] as shown in Figure 4, which are special cases of Figure 1. In case 1, IQL converges to the local optimum, and in case 2, IQL does not converge. Our I2Q can converge to the global optimum easily, in a fully decentralized way.

## 4.2 MPE

To investigate the effectiveness of I2Q in complex continuous environments with neural network implementation, we design a class of MPE-based differential games, where $N$ agents can move in the range $[-1, 1]$. In each timestep, agent $i$ acts the action $a_i \in [-1, 1]$, and the position of agent $i$ is updated as $x_i = \mathrm{clip}(x_i + 0.1 \times a_i, -1, 1)$ (*i.e.*, the updated position is clipped to $[-1, 1]$). The state is the position vector $\{x_1, x_2, \cdots, x_N\}$. The reward function of each timestep is defined as

$$r = \begin{cases} 0.5\cos(l\pi/m) + 0.5 & \text{if } l \le m \\ 0 & \text{if } m < l \le 0.6 \\ 0.15\cos(5\pi(l - 0.8)) + 0.15 & \text{if } 0.6 < l \le 1.0 \\ 0 & \text{if } l > 1.0 \end{cases}, \quad l = \sqrt{\frac{2}{N}\sum_{i=0}^{N} x_i^2}, \quad m = 0.13(N - 1).$$

The visualization of reward function of two-agent case is shown in Figure 5a. There is only one global optimum ($l = 0$ and $r = 1$) but infinite sub-optima ($l = 0.8$ and $r = 0.3$), and the region with $r > 0.3$ is very narrow and surrounded by the region with $r = 0$. So it is hard to learn the optimal policies in a decentralized way. The episode contains 100 timesteps, and the agents' positions are randomly initialized at the beginning of each episode. To verify the scalability, we test I2Q in the settings with different agent numbers $N$, and train the agents for eight random seeds in each setting. The results are shown in Figure 6. IDDPG always falls into local optimum due to the non-stationarity and outdated transition probabilities. To be optimistic towards the value punishment is an important technique in decentralized MARL. Although adopting the optimistic update, Hysteretic IDDPG [16] still falls into local optimum due to the non-stationarity problem. D3G [6] applies the idea

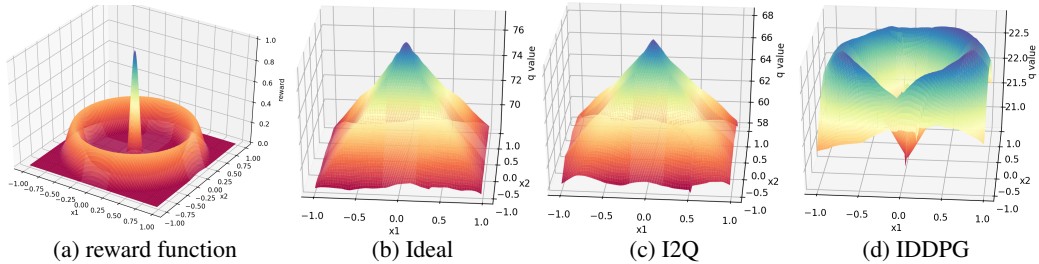

(a) reward function      (b) Ideal      (c) I2Q      (d) IDDPG

Figure 5: (a): Visualization of reward function in differential game with two agents. (b)-(d): Visualizations of learned values of agent 1 in differential game with two agents. x1-axis and x2-axis: agents 1 and 2's positions, respectively.

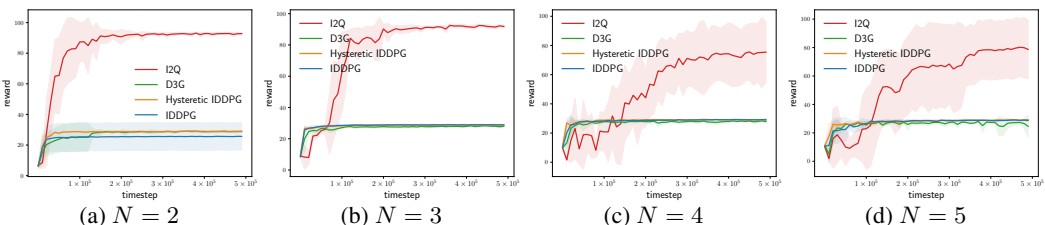

(a) $N = 2$      (b) $N = 3$      (c) $N = 4$      (d) $N = 5$

Figure 6: Learning curves on MPE-based differential games with different agent numbers $N$.

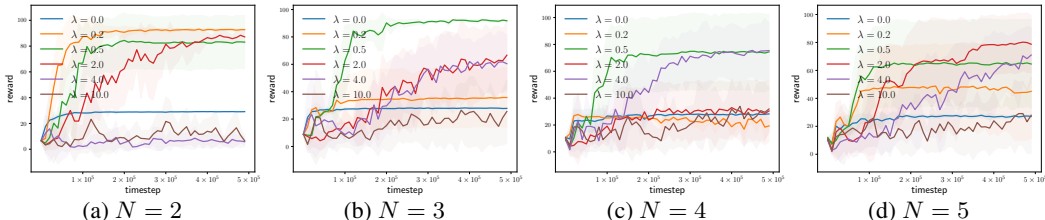

(a) $N = 2$      (b) $N = 3$      (c) $N = 4$      (d) $N = 5$

Figure 7: Learning curves on MPE-based differential games with different $\lambda$.

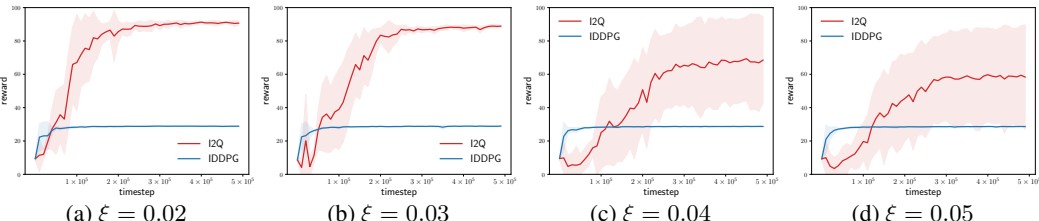

(a) $\xi = 0.02$      (b) $\xi = 0.03$      (c) $\xi = 0.04$      (d) $\xi = 0.05$

Figure 8: Learning curves on MPE-based differential games with different noise $\xi$.

of QSS in single-agent learning. To extend D3G into decentralized MARL, we train each agent independently using D3G, without the information of other agents. However, D3G cannot learn the optimal solution either, since the forward model in D3G requires the transition probabilities $P_i$ to be deterministic. However, when learning on the decentralized replay buffer, $P_i$ is the average of non-stationary transition probabilities and cannot be deterministic. Thus, D3G is not applicable to decentralized multi-agent learning. Learning on the ideal transition function inferred from $Q_i^{\mathrm{ss}}(s, s')$, I2Q can escape from local optimum and significantly outperform others. To thoroughly demonstrate the effectiveness of I2Q, in Figure 5, we visualize the learned values of agent 1 in the two-agent case. In Figure 5b, we let the agent 2 always act the optimal policy (moving to the position $x_2 = 0$), which means the agent 1 performs Q-learning on the true ideal transition probabilities. Thus, Figure 5b shows the optimal values under ideal transition probabilities. In Figure 5c, both agents use I2Q. I2Q values are very similar to the true ideal values, where the value of the center point ($l = 0$) is the highest, which means that the modeled transition function in I2Q is close to the true one. In Theorem 4 (see Appendix A), we theoretically analyze that the difference between ideal values (Figure 5b) and I2Q values (Figure 5c) is bounded by the model error. In Figure 5d, both agents use IQL, resulting in that the value of the center point ($l = 0$) is the lowest and the suboptimal points ($l = 0.8$) have the highest values.

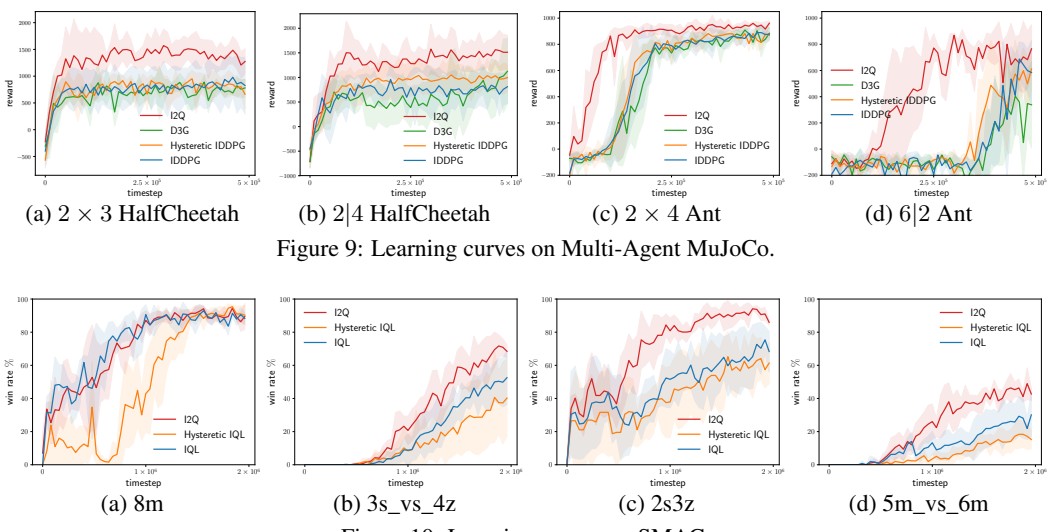

Figure 9: Learning curves on Multi-Agent MuJoCo.

Figure 10: Learning curves on SMAC.

To model the ideal transition function, next states predicted by $f_i$ should have the highest $Q_i^{\text{ss}}$ values (optimality) and be in the neighboring state set given state-action pair (referred to as neighborhood constraint). To achieve this objective, $\lambda$ in (12) controls the balance between optimality and neighborhood constraint. Figure 7 shows the effect of $\lambda$. When $\lambda$ is too small, the next state predicted by $f_i$ cannot maximize $Q_i^{\text{ss}}(s, s')$, and thus I2Q gets stuck at local optimum. Especially, when $\lambda = 0$, I2Q degrades into a popular model-based method MBPO [9]. When $\lambda$ is too large, *e.g.*, $\lambda = 10$, paying much more attention to maximize $Q_i^{\text{ss}}(s, s')$, $f_i$ would generate out-of-neighborhood states, causing that I2Q even cannot converge to local optimum. We notice that $\lambda = 0.5$ can generally achieve a good performance in different settings of $N$.

Although we assume deterministic environments in theoretical analysis, I2Q can also obtain performance gain in stochastic environments. To measure the impact of stochastic environments, we add a noise to the position update: $x_i = \text{clip}(x_i + 0.1 \times a_i + \xi \times z, -1, 1)$, where $z$ is a uniform random variable in $[-1, 1]$, and $\xi$ is a constant controlling the randomness. Figure 8 shows the performance with different $\xi$ in the three-agent case. In the stochastic environments, IDDPG still converges to local optimum, while I2Q always escapes from local optimum when $\xi = 0.02$ and $0.03$, free from the impact of stochastic environments. When $\xi = 0.04$ and $0.05$, the effect of stochasticity is nearly half of the effect of action. I2Q cannot always learn optimal policies with the strong stochasticity but still significantly outperforms IDDPG.

### 4.3 Multi-Agent MuJoCo and SMAC

To investigate the effectiveness of I2Q in *partially observable* environments, we perform experiments on Multi-Agent MuJoCo [18], where each agent independently controls one or some joints of the robot and could only observe the state of its own joints and bodies (with the parameter agent_obsk = 0). The results are shown in Figure 9 with eight random seeds. I2Q achieves higher rewards than IDDPG, which indicates that I2Q could be extended to partially observable tasks and obtain performance gain by performing Q-learning on ideal transition probabilities instead of transition probabilities in replay buffer. In the partially observable setting, we only consider two-agent cases. When there are more agents, each agent can only control one or two joints, and the observation range is too limited to learn strong policies. In Appendix B.4, we provide the 6-agent Walker and 8-agent Ant experiments with the full observation setting to verify the scalability.

We test I2Q on *partially observable and stochastic* SMAC tasks [23] with the version SC2.4.10, including both easy and hard maps [33]. We adopt the implementation of PyMARL [23] that takes as input the trajectory of partial observations, which is high-dimensional and not differentiable on some dimensions, meaning that directly optimizing (12) is hard. So the QSS model $Q_i^{\text{ss}}$ is not built on the trajectory of partial observations, but on the hidden state of $Q_i$ (the output of RNN layer), which is 64-dimension and differentiable, and the transition function $f_i$ predicts the next hidden state. We train the agents for four random seeds. D3G can only be used in continuous action space, so we do

not compare with it in SMAC. As shown in Figure 10, I2Q is capable of handling high-dimensional and stochastic tasks.

## 5 Closing Remarks

In this paper, we propose I2Q, a novel variant of independent Q-learning, to deal with the non-stationarity in decentralized cooperative MARL. I2Q models the ideal transition function, which depends on the optimal conditional joint policy of other agents and performs independent Q-learning on the modeled ideal transition function. Theoretically, we prove that I2Q converges to the optimal joint policy in deterministic environments. Empirically, I2Q obtains performance gain in a variety of multi-agent tasks.

*We focus on the base algorithm for fully decentralized learning in cooperative MARL, as it lay the algorithmic foundation for decentralized learning. Although we did not consider explicit coordination among agents, in partially observable environments coordination such as communication is desirable to improve empirical performance. We envision that more methods can be further built on the base algorithm. Moreover, currently, although fully decentralized learning algorithms may not empirically perform as well as CTDE methods like QMIX [22] in some benchmarks, e.g., hard maps in SMAC, we believe more efforts should be made by the MARL community on decentralized learning, considering its abundant benefits over CTDE. First, fully decentralized algorithms do not require a centralized learner during training, hence they are more applicable and cleaner, easier to implement [10]. Second, fully decentralized algorithms are more versatile as the individual learners are indifferent to the number of other agents, thus they have better scalability [34]. Third, fully decentralized algorithms make agents more robust to the presence of agents they were not trained with (such as humans) [30]. Therefore, fully decentralized learning is a fundamental problem in cooperative MARL, but remains to be investigated.*

## Acknowledgments and Disclosure of Funding

This work was supported by NSF China under grant 61872009. The authors would like to thank the anonymous reviewers for their valuable comments.

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
