# A Proof

**Theorem 2.** *In deterministic environments, all agents will converge to the optimal policies, if each agent $i$ performs Q-learning on the transition function $s'^* = \arg\max_{s' \in \mathcal{N}(s,a_i)} Q_i^{\text{ss}}(s,s')$.*

*Proof.* According to Theorem 1, we know that if each agent $i$ performs Q-learning on ideal transition function, all agents will converge to the global optimal policies. Therefore, to prove Theorem 2 is to prove that $s'^* = \arg\max_{s' \in \mathcal{N}(s,a_i)} Q_i^{\text{ss}}(s,s')$ is a transition function under one of the optimal conditional joint policies of other agents $\boldsymbol{\pi}^*_{-i}(s,a_i)$. Since

$$Q(s, \boldsymbol{a}) = Q(s, a_i, \boldsymbol{a}_{-i}) = r(s, s') + \gamma \max_{\boldsymbol{a}'} Q(s', \boldsymbol{a}'),$$

where $s'$ is the deterministic next state of $\boldsymbol{a}_{-i}$ when given $s$ and $a_i$, according to (10), we have

$$Q(s, a_i, \boldsymbol{a}_{-i}) = r(s, s') + \gamma \max_{\boldsymbol{a}'} Q(s', \boldsymbol{a}') = r(s, s') + \gamma \max_{s''} Q_i^{\text{ss}}(s', s'') = Q_i^{\text{ss}}(s, s').$$

Thus, $\max_{s'} Q_i^{\text{ss}}(s, s') = \max_{\boldsymbol{a}_{-i}} Q(s, a_i, \boldsymbol{a}_{-i})$, which means $Q_i^{\text{ss}}(s, s'^*) = Q(s, a_i, \boldsymbol{\pi}^*_{-i}(s, a_i))$. Therefore, when there are multiple $s'^* = \arg\max_{s' \in \mathcal{N}(s,a_i)} Q_i^{\text{ss}}(s,s')$, any one of them is an ideal transition function under one of the optimal conditional joint policies of other agents, and Theorem 2 holds. $\qquad\square$

**Theorem 3.** $\left\| Q_i^{sto} - Q_i \right\|_\infty \leq \frac{\Delta r}{1-\gamma} \left\| \frac{1 - P_{\text{env}}\left(s'^*|s,a_i,\boldsymbol{\pi}^*_{-i}(s,a_i)\right)}{1 - \gamma P_{\text{env}}\left(s'^*|s,a_i,\boldsymbol{\pi}^*_{-i}(s,a_i)\right)} \right\|_\infty$, *where* $\Delta r = r_{\max} - r_{\min}$.

*Proof.*

$\left\| Q_i^{sto} - Q_i \right\|_\infty$

$= \max\limits_{s,a_i} \left| \sum\limits_{s'} P_{\text{env}}\left(s'|s,a_i,\boldsymbol{\pi}^*_{-i}(s,a_i)\right) \left[ r(s,s') + \gamma \max\limits_{a_i'} Q_i^{sto}(s',a_i') \right] - r(s,s'^*) - \gamma \max\limits_{a_i'} Q_i(s'^*,a_i') \right|$

$\leq \max\limits_{s,a_i} \sum\limits_{s' \neq s'^*} P_{\text{env}}\left(s'|s,a_i,\boldsymbol{\pi}^*_{-i}(s,a_i)\right) \left| r(s,s') - r(s,s'^*) \right|$

$\quad + \gamma \sum\limits_{s' \neq s'^*} P_{\text{env}}\left(s'|s,a_i,\boldsymbol{\pi}^*_{-i}(s,a_i)\right) \left| \max\limits_{a_i'} Q_i^{sto}(s',a_i') - \max\limits_{a_i'} Q_i(s'^*,a_i') \right|$

$\quad + \gamma P_{\text{env}}\left(s'^*|s,a_i,\boldsymbol{\pi}^*_{-i}(s,a_i)\right) \max\limits_{a_i'} \left| Q_i^{sto}(s'^*,a_i') - Q_i(s'^*,a_i') \right|$

$\leq \max\limits_{s,a_i} \left(1 - P_{\text{env}}\left(s'^*|s,a_i,\boldsymbol{\pi}^*_{-i}(s,a_i)\right)\right) \left(\Delta r + \gamma \frac{\Delta r}{1-\gamma}\right) + \gamma P_{\text{env}}\left(s'^*|s,a_i,\boldsymbol{\pi}^*_{-i}(s,a_i)\right) \left\| Q_i^{sto} - Q_i \right\|_\infty.$

Therefore, we have

$$\left\| Q_i^{sto} - Q_i \right\|_\infty \leq \frac{\Delta r}{1-\gamma} \left\| \frac{1 - P_{\text{env}}\left(s'^*|s,a_i,\boldsymbol{\pi}^*_{-i}(s,a_i)\right)}{1 - \gamma P_{\text{env}}\left(s'^*|s,a_i,\boldsymbol{\pi}^*_{-i}(s,a_i)\right)} \right\|_\infty.$$

$\qquad\square$

The bound in Theorem 3 show that $Q_i$ of I2Q is closer to the true value if the transition probability of $s'^*$ is higher in stochastic environments. Since we sample transitions from $\mathcal{D}_i$ to update $f_i$ by maximizing (12):

$$\mathbb{E}_{s,a_i,s' \sim \mathcal{D}_i} \left[ \lambda Q_i^{\text{ss}}(s, f_i(s,a_i)) - (f_i(s,a_i) - s')^2 \right],$$

the second term makes the predicted next state $f_i(s,a_i)$ be close to the **high-frequency** next states given $s$ and $a_i$ in replay buffer $\mathcal{D}_i$, which means that the transition probability of the predicted next state $P_{\text{env}}\left(f_i(s,a_i)|s,a_i,\boldsymbol{\pi}^*_{-i}(s,a_i)\right)$ would not be too small. Thus, $Q_i$ of I2Q will be close to the true value and the worst cases where the predicted next states have very small transition probabilities can be avoided according to our implementation (12). ***That is the reason why I2Q can be successfully applied in stochastic environments***, as shown in Figure 8 and Figure 10.

**Theorem 4.** $Q_i^o(s, a_i)$ *is the value function learned on the ideal transition function* $s'^o = f_i^o(s, a_i)$, $Q_i(s, a_i)$ *is the value function learned on the modeled ideal transition function* $s' = f_i(s, a_i)$, *considering the continues state space,* $\|Q_i^o - Q_i\|_\infty \leq \frac{T + \gamma G}{1 - \gamma} \|f_i^o - f_i\|_\infty$, *where* $G = \max_{s,a} |\frac{\partial Q_i(s,a)}{\partial s}|, T = \max_{s,s'} |\frac{\partial r(s,s')}{\partial s'}|$, *when* $f_i(s, a_i)$ *is close to* $f_i^o(s, a_i)$.

*Proof.*

$\|Q_i^o - Q_i\|_\infty$

$$= \max_{s,a} \left| r(s, s'^o) + \gamma \max_{a_i'} Q_i^o(s'^o, a_i') - r(s, s') - \gamma \max_{a_i'} Q_i(s', a_i') \right|$$

$$\leq \max_{s,a} \left| r(s, s'^o) - r(s, s') \right| + \gamma \left| \max_{a_i'} Q_i^o(s'^o, a_i') - \max_{a_i'} Q_i(s', a_i') \right|$$

$$= \max_{s,a} \left| \frac{\partial r(s, s'^o)}{s'^o}(s' - s'^o) + o(s' - s'^o) \right|$$

$$+ \gamma \left| \max_{a_i'} Q_i^o(s'^o, a_i') - \max_{a_i'} [Q_i(s'^o, a_i') - \frac{\partial Q_i(s'^o, a_i')}{s'^o}(s' - s'^o) - o(s' - s'^o)] \right|$$

$$\leq \max_{s,s'} \left| \frac{\partial r(s, s')}{\partial s'} \right| \max |s' - s'^o| + \gamma \max_{s, a_i} |Q_i^o(s, a_i) - Q_i(s, a_i)| + \gamma \max_{s,a} |\frac{\partial Q_i(s, a)}{\partial s}| \max |s' - s'^o| + o(s' - s'^o)$$

$$= \gamma \|Q_i^o - Q_i\|_\infty + (T + \gamma G) \|f_i^o - f_i\|_\infty + o(\|f_i^o - f_i\|_\infty).$$

Omiting the remainder term, we have

$$\|Q_i^o - Q_i\|_\infty \leq \frac{T + \gamma G}{1 - \gamma} \|f_i^o - f_i\|_\infty.$$

$\square$

Theorem 4 shows that the difference between ideal values $Q_i^o$ and learned values $Q_i$ is bounded by the error of modeled transition function. As shown in Figure 5, the learned values of I2Q are very close to the ideal values, because I2Q accurately models the ideal transition function.

## B  Additional Results

### B.1  Hyperparameter $\lambda$

In Section 4.2, we have shown the effectiveness of $\lambda$. Here, we further discuss the cases where $\lambda = 0$. When $\lambda = 0$, I2Q degenerates into a popular model-based method MBPO [9]. In existing model-based methods, the modeled transition functions follow the transition probabilities in the replay buffer, which are non-stationary and outdated, so existing model-based methods also face the non-stationary problem. As shown in Figure 7 and Figure 11, I2Q outperforms I2Q with $\lambda = 0$, which verifies the effectiveness of the proposed objective function of the transition function $f_i$ in I2Q.

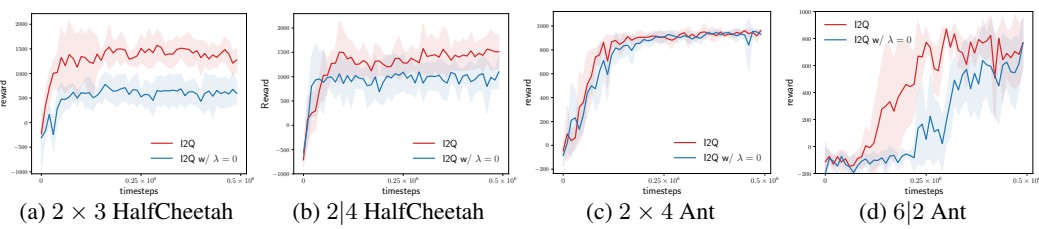

(a) $2 \times 3$ HalfCheetah    (b) 2|4 HalfCheetah    (c) $2 \times 4$ Ant    (d) 6|2 Ant

Figure 11: Learning curves with $\lambda = 0$.

### B.2  Two Update Rules

We propose two update rules for $Q_i$: rule 1 (14) and rule 2 (15). Figure 12 shows that the two rules achieve similar performance, especially in the tasks with long horizons where the reward of one timestep is insignificant compared with the cumulated return value. We use rule 2 in differential games and use rule 1 in SMAC and Multi-Agent MuJoCo.

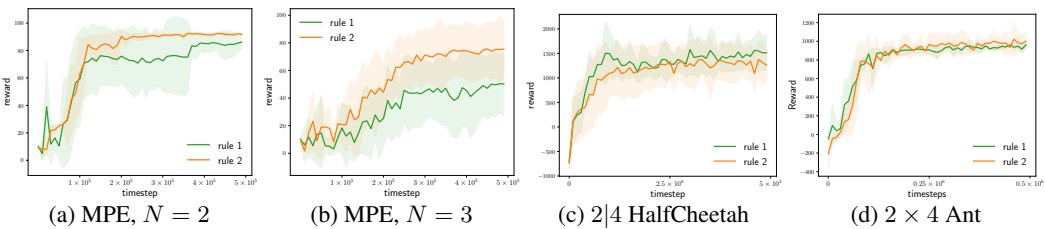

(a) MPE, $N = 2$     (b) MPE, $N = 3$     (c) 2|4 HalfCheetah     (d) $2 \times 4$ Ant

Figure 12: Learning curves of two update rules.

## B.3 Independent PPO

Independent PPO (IPPO) [4] is a strong on-policy decentralized MARL baseline. However, it is not fair to compare off-policy algorithms with on-policy algorithms, since on-policy algorithms do not use old data, which makes them weaker on sample efficiency [1], as shown in Figure 13.

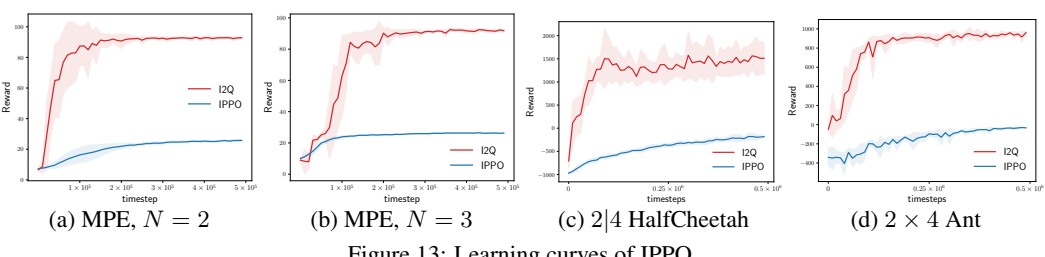

(a) MPE, $N = 2$     (b) MPE, $N = 3$     (c) 2|4 HalfCheetah     (d) $2 \times 4$ Ant

Figure 13: Learning curves of IPPO.

## B.4 Scalability

We test I2Q on 6-agent Walker and 8-agent Ant with full observation setting, the results are shown in Figure 14. Taking state information as input, IDDPG is strong enough, but I2Q can still obtain performance gain on 8-agent Ant.

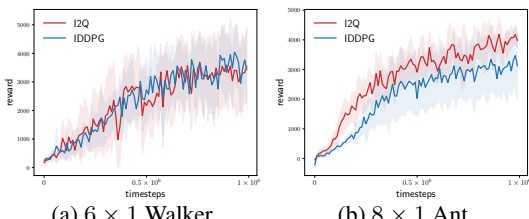

(a) $6 \times 1$ Walker     (b) $8 \times 1$ Ant

Figure 14: Learning curves on Multi-Agent MuJoCo.

## B.5 Multiple Optimal Joint Policies

In Section 3.4, we have analyzed that I2Q can easily solve the task with multiple optimal joint policies. Here, we give another way to solve this problem. When agent $i$ selects one of the optimal independent action, the neighboring state set $\mathcal{N}(s, a_i)$ contains one of the optimal next states, which have the max $Q_i^{\text{ss}}(s, s')$. The union of the neighboring state sets $\cap_i \mathcal{N}(s, a_i)$ is a singleton set, which contains the next state of the joint action. If each $\mathcal{N}(s, a_i)$ contains different optimal next states, the union $\cap_i \mathcal{N}(s, a_i)$ might contain none of the optimal ones. We let each agent independently and randomly initialize a same fixed noise function $\eta(s)$, using the same random seed, as a partial order, and makes decisions by the partial order

$$\arg\max_{a_i^*} \eta(s'^*), \quad s'^* = \arg\max_{s' \in N(s, a_i^*)} Q_i^{\text{ss}}(s, s').$$

Thus, each $\mathcal{N}(s, a_i)$ will contain the optimal next state with the max $\eta$ value, which is the next state of the joint action. Thus, the agents are coordinated by $\eta$. Since the noise function $\eta$ is randomly initialized, it is a small probability event to find multiple next states with the same $\eta$ value. We could also adopt other partial orders, *e.g.*, $l_1$ norm of $s'$.

We test I2Q on a one-stage matrix game with two optimal joint policies $(1, 2)$ and $(2, 1)$, as shown in Figure 15. If the agents independently select actions, they might choose the miscoordinated joint policies $(1, 1)$ and $(2, 2)$. IQL learns the suboptimal policy $(1, 1)$, but I2Q agents always select coordinated actions, though the value gap between the optimal policy and suboptimal policy is so small.

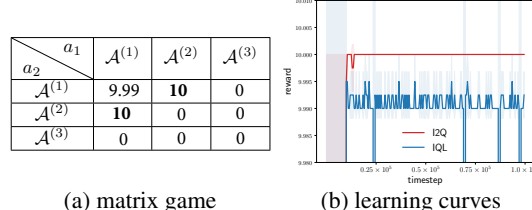

| $a_1$ 
 $a_2$ | $\mathcal{A}^{(1)}$ | $\mathcal{A}^{(2)}$ | $\mathcal{A}^{(3)}$ |
|---|---|---|---|
| $\mathcal{A}^{(1)}$ | 9.99 | **10** | 0 |
| $\mathcal{A}^{(2)}$ | **10** | 0 | 0 |
| $\mathcal{A}^{(3)}$ | 0 | 0 | 0 |

(a) matrix game        (b) learning curves

Figure 15: Learning curves on a one-stage matrix game with multiple optimal joint policies.

## B.6 Discussion on D3G

D3G predicts the optimal next state $s'^*$ with the highest QSS value and decodes actions in execution by training a inverse model $a_i = g(s, s'^*)$. Different from D3G, I2Q builds transition function $f_i(s, a_i)$ using QSS value and trains Q-learning on $f_i(s, a_i)$. Several drawbacks limit the performance of D3G in decentralized MARL. First, over-generalization of $g(s, s'^*)$ would lead to wrong $a_i$. Considering two inverse maps $(s, s'^1) \rightarrow a^1$ and $(s, s'^2) \rightarrow a^2$, where $s'^1$ and $s'^2$ are very similar but $a^1$ and $a^2$ are very different, which is common in complex continuous environments, trained by supervised learning, the inverse model $g$ may *mistakenly* predict $g(s, s'^1)$ as the average of $a^1$ and $a^2$, due to the generalization of neural network on state space. Since $a^1$ and $a^2$ are very different, the prediction error is large and leads to poor performance. Experimentally, we find it is hard to perfectly train the inverse model $g(s, s')$, and the prediction error of actions greatly influences the performance in practice. I2Q could avoid this problem since it learns accurate values $Q_i(s, a^1)$ and $Q_i(s, a^2)$ and selects the action with the highest value. Second, as stated in the experiment section, the implementation of D3G adopts a forward model, which requires the transition probabilities in replay buffer $\mathcal{D}_i$ to be deterministic, which is impossible in decentralized MARL. The motivation and implementation of I2Q are very different from that of D3G, which makes I2Q more suitable and practical in decentralized MARL, as shown in Figure 6 and Figure 9. The original D3G can only be used in continuous action space, and we extend it to discrete space and compare it in SMAC. Due to the analyzed drawbacks, discrete D3G cannot obtain a winning rate in SMAC, as shown in Table 1.

Table 1: Winning rate on SMAC.

|  | 8m | 3s_vs_4z | 2s3z | 5m_vs_6m |
|---|---|---|---|---|
| I2Q | 89% | 68% | 85% | 42% |
| discrete D3G | 0 | 0 | 0 | 0 |

## B.7 I2Q without Forward Model

For practicability, we design an approximation of I2Q without forward model $f_i$, which is enlightened by implicit Q-learning[12]. Implicit Q-learning learns the optimal value without predicting next actions, similarly, we can obtain QSS value without predicting next states. Following Implicit Q-learning, we introduce $V_i(s)$ and utilize expectile regression. Specifically, we update $Q_i^{\text{ss}}$ by minimizing

$$\mathbb{E}_{s,s',r \sim \mathcal{D}_i}[(Q_i^{\text{ss}}(s, s') - r - \gamma V_i(s'))^2],$$

update $V_i(s')$ by minimizing expectile loss

$$\mathbb{E}_{s,s' \sim \mathcal{D}_i}[L_2^\tau(Q_i^{\text{ss}}(s, s') - V_i(s'))], \quad L_2^\tau(u) = |\tau - 1(u < 0)|u^2.$$

Using the approximate QSS value, we have

$$Q_i^{\text{ss}}(s, s'^*) = \max_{s' \in \mathcal{N}(s, a_i)} Q_i^{\text{ss}}(s, s')$$

According to Eq 15 and the mathematical derivation between line 161 and line 162, we update $Q_i(s, a_i)$ to be $Q_i^{\mathrm{ss}}(s, s'^*)$ by minimizing

$$\mathbb{E}_{s, a_i, s' \sim \mathcal{D}_i}[(Q_i(s, a_i) - \max(\bar{Q}_i(s, a_i), \bar{Q}_i^{\mathrm{ss}}(s, s')))^2].$$

Although QSS value is a biased estimation in this implementation, the implementation without forward model is practical. We test I2Q w/o forward model in SMAC. The results are shown in Figure 16. I2Q w/o f shows similar performance with I2Q, and does not need to learn the forward model, thus it would be more practical in complex environments.

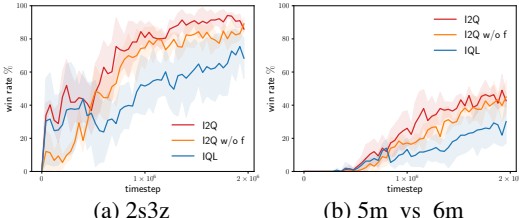

(a) 2s3z        (b) 5m_vs_6m

Figure 16: Learning curves on SMAC.

## C   Hyperparameters

In $2 \times 3$ HalfCheetah, there are two agents and each of them controls 3 joints of HalfCheetah. In $2|4$ HalfCheetah, there are two agents. One of them controls 2 joints, and one of them controls 4 joints. And so on.

In MPE-based (MIT license) differential games and Multi-Agent MuJoCo (MIT license), we adopt the implementation of SpinningUp [1] (MIT license), the SOTA implementation of DDPG, and follow all hyperparameters in SpinningUp. The discount factor $\gamma = 0.99$, the learning rate is $0.001$ with Adam optimizer, the batch size is $100$, the replay buffer contains $1 \times 10^6$ transitions, the hidden units are $256$.

In SMAC (MIT license), we adopt the implementation of PyMARL [23] (Apache-2.0 license) and follow all hyperparameters in PyMARL. The discount factor $\gamma = 0.99$, the learning rate is $0.0005$ with RMSprop optimizer, the batch size is $32$ episodes, the replay buffer contains $5000$ episodes, the hidden units are $64$. We adopt the version SC2.4.10.

The hyperparameter $\lambda$ has been fully tested in MPE-based differential games. In Multi-Agent MuJoCo, we set $\lambda = 0.01$, and in SMAC, we set $\lambda = 0.05$.

The experiments are carried out on Intel i7-8700 CPU and NVIDIA GTX 1080Ti GPU. The training of each MPE and Multi-Agent MuJoCo task could be finished in 5 hours, and the training of each SMAC task could be finished in 20 hours.

The code is available at https://github.com/jiechuanjiang/I2Q.