# OpenReview forum: "I2Q: A Fully Decentralized Q-Learning Algorithm"
_NeurIPS.cc/2022/Conference — NeurIPS 2022 Accept_

### Official Review · Reviewer_zsxJ · 2022-06-23

**Rating:** 6
**Confidence:** 4
**Soundness:** 3 good
**Presentation:** 4 excellent
**Contribution:** 3 good

**Summary:**

This paper presents I2Q, an algorithmic approach for decentralized MARL. The authors present the non-stationarity problem in this setting and propose to use "ideal transition probabilities" to solve it. Particularly, these are transition probabilities for which all agents are ensured to converge to an optimal solution when trained in a decentralized manner. The authors then propose to use the next state (in deterministic environments) as a representation of an action, and show that it induces an ideal transition probability, which ensures convergence to an optimal solution. They experiment on many baselines in various domains, showing the benefit of their approach.

**Questions:**

Most of my questions relate to things I already mentioned above.
1. Can stochasticity be addressed with "ideal transition probabilities" more explicitly? How can we learn such probabilities? How would this affect training? If not possible, what are the limitations?
2. How does approximation of $f$ affect convergence?

**Ethics Review Area:**

["I don’t know"]

**Limitations:**

The authors discuss limitations of their work. Some of these limitations coincide with points I've already raised. As mentioned above, I believe some of these points should be addressed more thoroughly in the paper.

**Strengths And Weaknesses:**

The paper proposes an elegant solution to the non-stationarity problem of decentralized MARL. I'm not able to say if it is the first method to solve this problem, and I hope one of the other reviewers will address this. The paper is clearly written, and the presentation is great. Also, I found everything to be easy to read and follow. Finally, the experiments section seems to have chosen a wide variety of tasks, and I'm glad the authors also chose to show results on the high dimensional problem of SCII.

----------------------------------------------------------------------------------------------------------

The paper doesn't have strong flaws, but there are some issues that make it a borderline paper for Neurips.

First, the theory is not very deep. There are many questions that remain open that the authors don't address theoretically, and I think are important for a better understanding of the problem. One of these, is convergence proof of I2Q, which the authors don't really prove, but only discuss informally.

Second, I feel that the deterministic assumption in the paper is a strong one, unless carefully addressed. In favor of the authors, they do discuss this in the paper, showing a result of the value gap, and also experiments on a wide variety of tasks. Still, I believe this is not adequately addressed. A stronger result for stochastic environments should be provided. I assume there exist some "ideal transition probabilities" for this setting. If it is the case that such are impossible to theoretically find, then this is an important point to address in the paper. Overall, I find Theorem 3 to be a trivial result. I wish to see an approach that tackles stochasticity explicitly, and provides a tighter bound for approximation errors.

Third, the fact that I2Q must learn a forward model is troubling, as model-based methods usually fail againt state of the art model-free methods on high dimensional tasks (unless latent spaces are used, such as in MuZero). The authors don't address the problem of estimating $f$ in their work. Moreover, I feel that this is not addressed fully in the experiments either.

Finally, while the experiments show results on different types of environments, I find that I2Q was not compared against enough baselines. There are a lot of new baselines on MARL, and particularly I would expect the authors to compare I2Q to at least three more baselines which are considered SOTA, and not only IQL - even if they are not decentralized.

----------------------------------------------------------------------------
Strengths:
1. A new solution for decentralized MARL
2. Proofs to formal statements seem correct
3. Paper is clearly written and presentation is great
4. Experiments show a variety of interesting tasks

Weaknesses:
1. Theory is weak
2. Stochastic environments should be addressed
3. Forward model should be addressed theoretically and in experiments
4. Experiments are lacking comparison to other algorithms

---

> ### Author Response · Authors · 2022-08-01
> **Other concerns**
>
> > "Informal" proof
>
> The reason why the proofs seem informal is that they are not too difficult, without involving much complex mathematical derivation. In Theorem 1, we prove that by learning on the ideal transition probabilities the agents could converge to the optimum, and in Theorem 2, we prove that Eq 11 is an ideal transition function. The overall logic is clear and rigorous.
>
> > Comparison to other baselines
>
> The existing studies of decentralized MARL are limited, and we do not find other recent papers for fully decentralized MARL *without* communication, which we think should be an advantage rather than a weakness of I2Q. Following the comments of Reviewer 8HEx, we also compare I2Q with independent SAC and TD3 in Appendix B10. It is **unfair** to compare I2Q with CTDE methods, which use the information of other agents. Although we do not run CTDE methods, since we use standard benchmarks, the results can be compared with that in published papers. In SMAC, to be honest, I2Q cannot outperform CTDE methods, e.g., VDN, QMIX, and QPLEX, where the winning rate could reach 100% in many tasks [1]. However, this cannot weaken the contribution of I2Q, because I2Q only uses local information. Due to the time limit, we will run the CTDE methods on SMAC and report the results in the final version.
>
> On the other hand, in matrix games and differential games (N=2,3), I2Q converges to global optimum and thus will not be inferior to any other baselines, including CTDE methods. In fact, **many CTDE methods cannot converge to the optimum on the two matrix games in Figure 4**, as shown in [1] (Figure 2 and Figure 6 of [1]), but I2Q can converge to the optimum easily. We will also run CTDE methods on matrix games in the final version.
>
> [1] Wang, Jianhao, et al. "QPLEX: Duplex Dueling Multi-Agent Q-Learning." *International Conference on Learning Representations*. 2021.

---

> > ### Comment · Reviewer_zsxJ · 2022-08-08
> > **Response**
> >
> > I thank the authors for their detailed rebuttal. While the authors responded to all my points, they remain points of weakness in the paper. Particularly as learning a stochastic model is hard and it is unclear how learning a stochastic model would perform in practice in this case. Also, I still find the theoretical aspects of this paper weak; in particular many of the results are not very informative and are better off moved to the appendix.
> >
> > Nevertheless, I find this paper to have other strong qualities, which make it a great candidate for Neurips. Therfore, I am in favor of accepting this paper.

---

> ### Author Response · Authors · 2022-08-01
> **Removing forward model**
>
> > Removing forward model
>
> Very insightful suggestion! First, as you have mentioned, latent space can help the model learning in high dimensional tasks, and we have claimed that in discrete state space or large state space, we can map the state space to a **continuous embedding space** (line 170), and apply I2Q on the embedding space, and in SMAC, $Q_i^{ss}$ and $f_i$ are built on the **hidden state** of $Q_i$ (the output of RNN layer) (line 296). And we also demonstrated the question "How does approximation of $f_i$ affect convergence?" in Theorem 4.
>
> Second, the forward model is also exactly what we want to remove. We **had designed a kind of I2Q implementation without forward model**, enlightened by implicit Q-learning [1], but did not include it in the original version because it is just an approximation. Implicit Q-learning learns the optimal value without predicting next actions, similarly, we can obtain QSS value without predicting next states. Following Implicit Q-learning, we introduce $V_i(s)$ and utilize expectile regression. Specifically, we
>
> update $Q_i^{ss}$ by minimizing $E_{s, s', r \sim D_i}[(Q_i^{ss}(s, s')-r-\gamma V_i(s'))^2]$
>
> update $V_i(s)$ by minimizing expectile loss $E_{s, s' \sim D_i}[L_2^{\tau}(Q_i^{ss}(s, s')-V_i(s'))]$, $L_2^{\tau}(u)=|\tau-1(u<0)| u^2$.
>
> Using the approximate QSS value, we have
>
> $$Q_i^{ss}(s, s'^*) = \max_{s' \in N(s,a_i)} Q_i^{ss}(s, s')$$
>
> According to Eq.15 and the mathematical derivation between line 161 and line 162, we update $Q_i(s, a_i)$ to be $Q_i^{ss}(s, s'^*)$ by minimizing $$E_{s, a_i,s' \sim D_i}[(Q_i(s, a_i)-\max(\bar{Q}_i(s, a_i), \bar{Q}_i^{ss}(s, s')))^2].$$
>
> Although QSS value is a biased estimate in this implementation, the implementation without forward model is practical. We test I2Q w/o forward model in SMAC. **The results are shown in Figure 19**. I2Q w/o f shows similar performance to I2Q, and does not need to learn the forward model, thus it would be more practical in complex environments.
>
> [1] Kostrikov, Ilya, Ashvin Nair, and Sergey Levine. "Offline Reinforcement Learning with Implicit Q-Learning." *International Conference on Learning Representations*. 2021.

---

> ### Author Response · Authors · 2022-08-01
> **Stochastic environments**
>
> > Stochastic environments
>
> Stochastic environment is really an important topic. To respond to the concerns, we will summarize the analysis in the original version, discuss the tighter bound of Theorem 3, provide a new extension of I2Q for stochastic environments, and perform experiments on other stochastic games.
>
> First, in the original version, we do many stochastic experiments, including stochastic matrix games in Appendix B.6, noisy differential games in Fig. 8, and SMAC. In stochastic matrix games and noisy differential games, we impose strong stochasticity, and I2Q still outperforms IQL.
>
> Second, in the original version, we provide **a tighter bound** in Appendix A, and now we have updated the tighter bound to the main pages in the revision. The bound is meaningful for explaining **why I2Q can be successfully applied in stochastic environments** (colored line in Appendix A). As discussed in Appendix A, since we update $f_i$ by Eq.12, the second term of Eq.12 makes the predicted next states $f_i(s, a_i)$ close to the **high-frequency** next states in the replay buffer, which means that the transition probability of $f_i(s, a_i)$ would not be too small. So I2Q value will be close to the true value and the worst cases where $f_i(s, a_i)$ has very small transition probabilities can be avoided.
>
> Third, **to *explicitly* model ideal transition probabilities in stochastic environments**, we propose a new extension of I2Q. **The key idea is to predict the transition probabilities, instead of the next state**. We extend $Q_i^{ss}(s,s')$ to $Q_i^{s\rho}(s,\rho(\cdot|s))$, the value of state and the probabilities of **all next states** given the state. Similar to Eq. 11, the ideal transition probability is
> $$\rho^*(s,a_i)= \arg \max_{\rho(s,a_i) \in P(s,a_i)}  Q_i^{s\rho}(s,\rho(s,a_i))$$
> $P(s,a_i)$ is the set of possible transition probabilities given $s$ and $a_i$. We extend $f_i(s,a_i)$ to predict $\rho^*(s,a_i)$, which could be a softmax or a Gaussian. Then we do not maintain a whole buffer $D_i$ but a sequence of small buffers $\{D_i^m\}$. In the collection of each buffer $D_i^m$, the agents act deterministic policies, and between different buffers, the agents update the policies. So, each $D_i^m$ contains a transition probability $\rho^m(\cdot|s)$ under deterministic policies. Then, at each update, we
>
> 1 randomly select a buffer $D_i^m$
>
> 2 update $Q_i^{s\rho}$ by minimizing $E_{s, s', r \sim D^m_i}[(Q_i^{s \rho}(s, \rho^m(\cdot|s))-r-\gamma \bar{Q}_i^{s \rho }(s', f_i(s', a'^*_i)))^2]$
>
> 3 Update $f_i$ by maximizing $E_{s, a_i \sim D_i^m}[\lambda Q_i^{s \rho}(s, f_i(s, a_i))-(f_i(s, a_i)-\rho^m(\cdot|s))^2]$
>
> 4 Update $Q_i$ by minimizing $E_{s, a_i \sim D_i^m}[(Q_i(s, a_i)-\bar{Q}_i^{s \rho}(s, f_i(s, a_i))^2].$
>
> The whole training process is similar to deterministic I2Q, just replacing $s'$ with $\rho$. For the objective of $f_i$, the first term guarantees the optimality and the second term enforces the predicted transition probabilities to be in the set $P(s,a_i)$. Following the proof of deterministic I2Q, we could similarly prove that $f_i$ learns the ideal transition probabilities in stochastic tasks and according to Theorem 1, the agents converge to the optimum.
>
> However, although stochastic I2Q theoretically finds ideal transition probabilities, predicting the transition probabilities in complex environments is much more difficult than predicting the next state. Since we have analyzed the reason why deterministic I2Q can be successfully applied in stochastic environments in Theorem 3, we still choose the deterministic version in our paper due to its **practicability**.
>
> Finally, we test stochastic I2Q proposed above in stochastic games with 3 agents, 30 states, infinite-horizon. The action space of each agent is 5. Each state will transition to any state given a joint action according to transition probabilities. The transition probabilities and reward function are randomly generated and fixed. We generate 20 games, and **normalized rewards are shown in Figure 18**. I2Q significantly outperforms IQL and is very close to the optimum, empirically verifying that stochastic I2Q could learn ideal transition probabilities in stochastic tasks.

---

### Official Review · Reviewer_8HEx · 2022-07-11

**Rating:** 5
**Confidence:** 3
**Soundness:** 3 good
**Presentation:** 2 fair
**Contribution:** 2 fair

**Summary:**

This paper proposes a new MARL algorithm under the DTDE paradigm. Specifically, the proposed algorithm (I2Q) is introduced based on the ideal transition probability (where each agent assumes that the others adopt the optimal actions for each decision) and a previous idea named QSS-Learning. Theoretical guarantee on the convergence of the proposed algorithm is provided under certain conditions. For experimental studies, the significant superiority of I2Q is demonstrated in matrix games, MPE, MA MuJoCo and SMAC.

**Questions:**

Questions:
+ If we add a fixed randomly initialized reward function and the performance tolerance to the other algorithms, is there a performance improvement?


I appreciate the authors’ reasonable introduction of QSS for DTDE.
However, my major concern is the lack of novelty in the proposed algorithm. The current algorithm seems to be a simple application of the QSS-Learning in the MARL scenario, the QSS and the prediction of next state in this paper are consistent with the original QSS paper.

&nbsp;

Suggestions:

+ I recommend the authors to combine the proposed approach with more base RL algorithms, such as PPO, SAC and TD3 to better evaluate of the generality of the method.

+ The authorss should add a discussion of QSS and the differences between I2Q and QSS in the related work or background.


**Limitations:**

The rationality and limitations of the main assumptions adopted in this paper are discussed in Sec.3.4.

**Strengths And Weaknesses:**

Pros:
+ This paper is clearly written.
+ The experimental part is relatively diverse and adequate.

&nbsp;

Cons:
- The novelty of the proposed algorithm is limited.

&nbsp;


Minor issues and typos:
- l.299 "SAMC" → "SMAC"

---

> ### Author Response · Authors · 2022-08-01
> **Response to Reviewer 8HEx**
>
> > Novelty and difference between I2Q and D3G (QSS)
>
> We think your main concerns are the novelty and difference between I2Q and D3G (QSS). The main novelty and contribution are not just introducing QSS-learning into MARL but proposing **a new paradigm** where agents independently learn on ideal transition probabilities. QSS is merely a technique for building the ideal transition probabilities, which can also be learned by other techniques. For example, in the response to Reviewer zsxJ(or Appendix B8), we extend $Q_i^{ss}(s,s')$ to $Q_i^{s\rho}(s,\rho(\cdot |s))$ (i.e., the value of state and the probabilities of all next states given the state), to build the ideal transition probabilities in stochastic environments. Researchers could follow this paradigm to design other methods to build ideal transition probabilities. Thus, we believe our work is novel.
>
> The difference between I2Q and D3G (QSS) has been discussed in Appendix B.7 of the original version. The motivation and implementation of I2Q are different from that of D3G, and **the main difference, which makes D3G not suitable for decentralized MARL**, is that the Cycle loss in D3G requires the transition probabilities in replay buffer $D_i$ to be deterministic. However, this is impossible even if the environment is deterministic, because other agents are also updating policies. Therefore, D3G still suffers from non-stationarity and cannot achieve competitive performance in the experiments.
>
> Besides adopting QSS-learning, we still analyze the reason why I2Q can be successfully applied in **stochastic environments**, and extend I2Q to the version **without forward model** in the response to Reviewer zsxJ(or Appendix B9), so we believe I2Q is a novel work instead of just a simple application of QSS-learning.
>
> > If we add a fixed randomly initialized reward function and the performance tolerance to the other algorithms, is there a performance improvement?
>
> The randomly initialized reward function is introduced to only remedy the assumption of only one optimal policy for our theoretical results. Empirically, this is not required. **In all experiments, we do not use a randomly initialized reward function for I2Q and other baselines, thus the comparison is fair.**
>
>
> > More base RL algorithms
>
> Thanks for your advice on generality! Since I2Q is a variant of Q-learning, it could be instantiated on Q-learning methods, e.g., DDPG, SAC, and TD3. We test I2Q on independent SAC and TD3, and **the results are shown in Figure 20**. I2Q also obtains performance gain on the two base algorithms. PPO is not a Q-learning method so it cannot be the base algorithm of I2Q.

---

### Official Review · Reviewer_fWps · 2022-07-12

**Rating:** 8
**Confidence:** 3
**Soundness:** 3 good
**Presentation:** 2 fair
**Contribution:** 3 good

**Summary:**

This paper presents an important and interesting approach on fully decentralized MARL. Fully decentralized Q-learning is highly applicable to realistic and real-world applications. The method is evaluated extensively, showing a great potential.

**Questions:**

- I like the idea behind this approach and I believe that this is a strong work with potential for real-world applicability.

- Some important related work on fully decentralized MARL are not covered [1-2]. These works and similar prior work must be discussed to cover similarities and differences so that readers can select the proper method based on their applications.

[1] Zhang, Kaiqing, et al. "Fully decentralized multi-agent reinforcement learning with networked agents." International Conference on Machine Learning. PMLR, 2018.

[2] Konan, Sachin G., Esmaeil Seraj, and Matthew Gombolay. "Iterated Reasoning with Mutual Information in Cooperative and Byzantine Decentralized Teaming." International Conference on Learning Representations. 2021.

- The approach potentially needs to be compared against SOTA fully decentralized MARL work to evaluate its utility and feasibility.

**Limitations:**

Yes

**Strengths And Weaknesses:**

## Strengths
- Theoretical analysis
- Extensive evaluations
- Interesting perspective to the MARL problem with potential real-world applicability

## Weaknesses
- missing out on some related work on fully decentralized MARL
- Lack of SOTA baselines

---

> ### Author Response · Authors · 2022-08-01
> **Response to Reviewer fWps**
>
> > About the two references
>
> Thanks for the missing references, but the two studies do not follow our fully decentralized settings in Section 3.1, where communication is fully disabled and agents cannot share any information. Both [1] and [2] **allow communication** with neighboring agents according to a time-varying communication network (Definition 2.1 in [1] and Problem Formulation in [2]). In [1], neural network parameters are shared between neighboring agents, and in [2], actions produced by communicative policies are shared between neighboring agents. So we think the two methods should be classified as *decentralized learning with communication*. We have summarized the difference in Related Work of the revision.
>
> > SOTA fully decentralized MARL work
>
> Fully decentralized MARL is a new field especially when combined with neural networks, and we do not find other recent papers for fully decentralized MARL without communication. So we believe our I2Q is a novel work in this field. In fact, IQL is a naive but widely adopted decentralized method and Hysteretic IQL is a classic decentralized baseline. And we compare the two methods and also IPPO in Appendix. Moreover, following the comments of Reviewer 8HEx, we also compare I2Q with independent SAC and TD3 in Appendix B10.
>
> [1] Zhang, Kaiqing, et al. "Fully decentralized multi-agent reinforcement learning with networked agents." International Conference on Machine Learning. PMLR, 2018.
>
> [2] Konan, Sachin G., Esmaeil Seraj, and Matthew Gombolay. "Iterated Reasoning with Mutual Information in Cooperative and Byzantine Decentralized Teaming." International Conference on Learning Representations. 2021.

---

> > ### Comment · Reviewer_fWps · 2022-08-03
> > **Respond to Authors' Response**
> >
> > Thank you! I am satisfied with the response provided to my review, and after reading the reviews by other reviewers and their respective responses, I'm increasing my score. This is a solid paper.

---

### Author Response · Authors · 2022-08-01
**Thanks for your suggestions!**

We thank all the reviewers for the efforts on reviewing our paper and the insightful suggestions! Following the comments, we extend I2Q, including I2Q for stochastic environments, I2Q w/o forward model, and instantiations on more base algorithms. We include these extensions and experimental results in Appendix B8, B9, and B10 in the revision (with colored section title), and answer all questions in detail. We hope our feedback could address the concerns and look forward to further discussions.

---

### Meta-Review · Area_Chair_qznN · 2022-08-24

**Recommendation:** Accept
**Confidence:** Less certain

**Metareview:**

The paper presents a novel method for dealing with nonstationarity in decentralized multi-agent reinforcement learning (MARL). While there are some concerns about the level of novelty, the approach is interesting and presented well. There are also concerns about the discussion and comparison with the state-of-the-art in decentralized MARL methods. We suggest the authors include comparisons to other decentralized MARL methods (such as the ones below) or state why such comparisons are not reasonable.

Omidshafiei, Shayegan, et al. "Deep decentralized multi-task multi-agent reinforcement learning under partial observability." International Conference on Machine Learning. PMLR, 2017.

Palmer, Gregory, et al. "Lenient Multi-Agent Deep Reinforcement Learning." Proceedings of the International Conference on Autonomous Agents and MultiAgent Systems. 2018.

Lyu, Xueguang, and Christopher Amato. "Likelihood Quantile Networks for Coordinating Multi-Agent Reinforcement Learning." Proceedings of the 19th International Conference on Autonomous Agents and MultiAgent Systems. 2020.

**Award:**

No

---

### Decision · Program_Chairs · 2022-09-14

Accept